# Replacing murine insulin 1 with human insulin protects NOD mice from diabetes

**Colleen M. Elso[1,2], Nicholas A. Scott●[1¤], Lina Mariana[1], Emma I. Masterman[1], Andrew P. R. Sutherland[1,2], Helen E. Thomas[1,2], Stuart I. Mannering●[1,2]***

**1** Immunology and Diabetes Unit, St. Vincent's Institute of Medical Research, Fitzroy, Victoria, Australia,
**2** Department of Medicine, University of Melbourne, St. Vincent's Hospital, Fitzroy, Victoria, Australia

¤ Current address: Lydia Becker Institute of Immunology and Inflammation, University of Manchester, Manchester, United Kingdom.
* smannering@svi.edu.au

**Data Availability Statement:** All relevant data are within the manuscript and its Supporting Information files.

## Abstract

Type 1, or autoimmune, diabetes is caused by the T-cell mediated destruction of the insulin-producing pancreatic beta cells. Non-obese diabetic (NOD) mice spontaneously develop autoimmune diabetes akin to human type 1 diabetes. For this reason, the NOD mouse has been the preeminent murine model for human type 1 diabetes research for several decades. However, humanized mouse models are highly sought after because they offer both the experimental tractability of a mouse model *and* the clinical relevance of human-based research. Autoimmune T-cell responses against insulin, and its precursor proinsulin, play central roles in the autoimmune responses against pancreatic beta cells in both humans and NOD mice. As a first step towards developing a murine model of the human autoimmune response against pancreatic beta cells we set out to replace the murine insulin 1 gene (*Ins1*) with the human insulin gene (*Ins*) using CRISPR/Cas9. Here we describe a NOD mouse strain that expresses human insulin in place of murine insulin 1, referred to as HuPI. HuPI mice express human insulin, and C-peptide, in their serum and pancreata and have normal glucose tolerance. Compared with wild type NOD mice, the incidence of diabetes is much lower in HuPI mice. Only 15–20% of HuPI mice developed diabetes after 300 days, compared to more than 60% of unmodified NOD mice. Immune-cell infiltration into the pancreatic islets of HuPI mice was not detectable at 100 days but was clearly evident by 300 days. This work highlights the feasibility of using CRISPR/Cas9 to create mouse models of human diseases that express proteins pivotal to the human disease. Furthermore, it reveals that even subtle changes in proinsulin protect NOD mice from diabetes.

## Introduction

Type 1 diabetes (T1D) is an autoimmune disease caused by the T-cell mediated destruction of the pancreatic, insulin-producing beta cells [1, 2]. This leads to a primary insulin deficiency and subsequent metabolic disease [3]. Since its discovery in 1980 [4], the Non-obese diabetic (NOD) mouse has been the preferred mouse model for research into human T1D.

**Funding:** This work was supported by a: A Millennium Award (Y17M1-MANS) from Diabetes Australia (SM) and a Diabetes Australia Project grant to CE (Y18G-ELSC); Australian National Health and Medical Research Council (GNT1123586) to SM and JDRF (JDRF 2-SRA-2018-568-S-B) to SM. The funders had no role in study design, data collection and analysis, decision to publish, or preparation of the manuscript.

**Competing interests:** The authors have declared that no competing interests exist.

Autoimmune diabetes in the NOD mouse shares many pathological features with human type 1 diabetes [5]. For example, diabetes develops spontaneously, the MHC II genes greatly impact upon susceptibility and immune-cell infiltration of the islets of Langerhans is observed [6].

Many autoantigens have been implicated in the immune pathogenesis of human T1D and NOD autoimmune diabetes [7, 8]. However, T cells specific for insulin, and its precursor pro-insulin, have been shown to cause diabetes in the NOD mouse [9–11]. Although pathogenesis cannot be measured directly in humans, T-cell responses to (pro)insulin are strongly implicated in human type 1 diabetes [12–15]. Humans carry one insulin gene (*INS*), whereas the mouse genome contains two nonallelic insulin genes, *Ins1* (chromosome 19) and *Ins2* (chromosome 7) [16]. *Ins1* is predominantly expressed in the beta cells and *Ins2* is expressed in both the thymus and beta cells [17–19]. Genetic deletion of the *Ins1* gene protects NOD mice from autoimmune diabetes [20], whereas *Ins2* knock-out NOD mice rapidly develop autoimmune diabetes [21]. These observations suggest that insulin 2 protects NOD from diabetes, perhaps by promoting the deletion of (pro)insulin specific T cells in the thymus, whereas insulin 1 is primarily a target of diabetes-causing T cells.

The NOD mouse is a powerful research tool, but the relevance of findings from mouse models to human type 1 diabetes remain uncertain [22, 23]. Several groups have attempted to develop mouse models that harbor human cells, or express human genes, in an effort to develop models that are both experimentally tractable and clinically relevant. Broadly, these models fall into two camps: (i) immune-deficient mice that are transplanted with human cells [24, 25] and (ii) NOD mice that have been genetically manipulated to express relevant human genes [26, 27]. Human cell transplant models vary depending upon the donor cells and must be reestablished for each experiment. Genetic models are more defined, but the expression of human transgenes does not always follow that of the murine orthologues [28]. Variability of expression arises because the integration of the transgene is random, and expression is subject to the regulatory environment into which it integrates. Transgenes may also integrate in tandem arrays leading to variable expression levels depending upon the number of copies of the transgene. Furthermore, to avoid functional impacts of the endogenous murine genes these also need to be disrupted.

The emergence of CRISPR/Cas9 technology provides an opportunity to edit the mammalian genome with unprecedented precision [29]. Here we asked if CRISPR/Cas 9 could be used to replace the coding sequence of murine insulin with human insulin. This would allow the expression of human insulin, in place of murine insulin 1, with minimal disruption of the murine genome. A NOD mouse that expresses human insulin from the murine insulin 1 locus would be an important step towards developing a NOD mouse model that mimics the human T-cell response to proinsulin [12, 14]. Here we report the generation of a human insulin replacement mouse at the murine insulin 1 locus. We show that this mouse produces human insulin, develops insulitis, but is largely protected from diabetes, similarly to *Ins1* knock-out NOD mice.

## Materials and methods

### Production of human insulin knock-in mice

All mouse experiments were approved by the Animal Ethics Committee of the St Vincent's Hospital Melbourne (AEC:019.14 and 020.14) and carried out under the NHMRC Code of Practice. NOD/Lt-*Ins1*$^{em1(INS)Tkay}$ (HuPI) mice were created at the Australian Phenomics Network (APN, Melbourne, Australia) using CRISPR-Cas9-mediated replacement of murine *Ins1* with human *INS*. sgRNAs were designed to target the 5' and 3' regions of *Ins1*. A homology-directed repair (HDR) template was produced by cloning into pUC19: 701bp 5' homology

arm, 333bp sequence encoding human INS and 506 bp 3' homology arm (resulting in 559bp homology at 3' end of the INS gene, see S1 Fig). NOD/Lt embryos were microinjected cytoplasmically with 30 ng/µl Cas9 mRNA, 15 ng/µl each sgRNA and 30 ng/µl linearized HDR template. In some experiments they were incubated in 50µM SCR7 (a non-homologous end joining (NHEJ) inhibitor [30]) overnight prior to transfer into recipients (S1 Table). The sequences and on- and off-target scores for the sgRNAs are shown (S1 Table). One of 51 pups contained the complete insert in the correct location (S2 Table). This pup arose from a micro-injection using sgRNAs 1 and 3 (S1 Table). This was confirmed by PCR and sequencing. The mutation was backcrossed to NOD/Lt for at least two, but for almost all experimental mice, three, generations and intercrossed to produce human insulin knock-in homozygous mice. No differences were detected in mice from different numbers of backcrosses.

## ELISAs for human insulin and C-peptide

Seven to twelve week old HuPI mice were fasted 5–6 h before collection of serum. Human insulin and C-peptide, and murine insulin were measured by ELISA (Mercodia, Uppsala, Sweden). Mouse insulin measurements were adjusted for cross-reactivity with human insulin as per manufacturer's instructions. Data are expressed as mean ± SEM and differences assessed using an unpaired Student's two-tailed t test.

## PCR

RNA was extracted from mouse pancreas samples using Nucleospin RNA extraction columns (Macheray Nagal) after mechanical disruption. On column DNase digestion was performed as per manufacturer's instructions. cDNA was made with Superscipt III (Invitrogen) using 400ng RNA and 2.5µM oligo-dT. Cloneamp HiFI Premix (Takara) was used to amplify cDNA with 0.2µM primers as per manufacturer's instructions. PCR conditions: 35 cycles of 98˚C 10 sec, annealing temp 15sec, 72˚C 10sec, followed by 72˚C 7min. Primer sequences and annealing temperatures can be found in S4 Table.

## Glucose tolerance test

A glucose tolerance test was carried out on 6-10-week-old HuPI mice. Mice were fasted for 6 h before intraperitoneal injection of glucose solution (2g/kg, Baxter, Deerfield, IL, USA), followed by measurement of blood glucose at 0, 15, 30, 45, 60, and 120 min (Accu-Chek Performa, Roche). A two-way ANOVA was performed to detect differences between the groups.

## Immunohistochemistry and immunofluorescence

Pancreata were fixed in 4% paraformaldehyde and embedded in paraffin. Pancreatic sections were stained with 4 µg/mL of anti-human proinsulin/C-peptide (GN-ID4, Developmental Studies Hybridoma Bank, IA, USA) or an isotype control (40BI3.2.1-s, Developmental Studies Hybridoma Bank, IA, USA) for 45 min at RT followed by rabbit anti-rat IgG-HRP (Dako) for 45 min at RT. Slides were then incubated for 1 min with 3,3'-Diaminobenzidine and counterstained with haematoxylin. No staining was seen in pancreas of wildtype NOD mice. For immunofluorescence, pancreata were frozen in OCT (optimal cutting temperature) compound, cryosectioned and fixed in acetone before staining with the anti-human proinsulin/C-peptide (GN-ID4) or isotype control (40BI3.2.1-s) antibodies, followed by staining with an anti-rat IgG-AF568.

## Diabetes incidence

Age-matched HuPI homozygotes, heterozygotes and wildtype controls were tested weekly for diabetes by measuring their urinary glucose concentrations (Diastix, Bayer) until 300d of age. Mice were declared diabetic if they had high urinary glucose readings for three consecutive days and a blood glucose reading >15mmol/L (Accu-Check Proforma, Roche). Pairwise comparisons of the diabetes incidence between mouse strains were performed using the log-rank test.

## Insulitis

Pancreata were collected from 100d old or 300d old mice, fixed in Bouin's fixative and embedded in paraffin. Sections (5 μm) were cut at three different levels 100 μm apart and stained with haematoxylin and eosin. Lymphocytic infiltration of the islets was scored as described [31], briefly: 0 = no infiltration, 1 = peri-insulitis, 2 = < 25% islet infiltrated, 3 = > 25% of the islet infiltrated and 4 = complete infiltration. To calculate the insulitis score, the number of islets in each scoring category: 0, 1, 2, 3, 4, was multiplied by 0, 0.25, 0.5, 0.75 and 1, respectively. This value was then divided by the total number of islets to provide a weighted average insulitis score for each mouse. Differences between strains were assessed using a one-way ANOVA.

# Results

## Production of human insulin knock-in mice

Murine INS1 is predominantly expressed in the beta cell, whereas INS2 is expressed both in the thymus and beta cell [18, 19]. To ensure expression of human insulin in the beta cells of knock-in mice, we replaced the amino acid coding region of mouse *Ins1* with the human *INS* coding sequence (Fig 1). Note that the amino acid sequence of the 16 amino acids at the COOH terminal end of insulin A-chain is identical between human and murine insulin, so the sequence of this region did not change (Fig 1). CRISPR-Cas9 was used to induce double strand DNA breaks close to the start and stop codons of *Ins1* (S1 Table). The human coding sequence was introduced by homologous recombination from a repair construct containing homologous regions flanking the *Ins1* coding region (Fig 1A and 1B). On initial screening, 5 of 51 founder mice contained the 3' end of the human insulin gene (S2 Table). Further analysis confirmed that one founder mouse contained the entire human insulin coding sequence in the correct genomic location. This founder mouse was bred to WT NOD and was established as a line (called HuPI). These mice are healthy, viable and display no gross abnormalities. Offspring of an F1 intercross carried the knock-in allele at expected Mendelian ratios (S3 Table).

## HuPI mice produce human insulin

The concentration of human insulin and C-peptide in HuPI serum was determined by ELISA (Fig 2A and 2B). Human insulin was detected in human insulin knock-in heterozygous mice, but higher concentrations of insulin (KI/+; 0.4–3.8 mU/L v KI/KI; 2.3–17.2 mU/L) and C-peptide (KI/+; 0–292 pmol/L v KI/KI; 28–556 pmol/L) were observed in human insulin homozygous knock-in mice. As expected, both human insulin and murine insulin could be detected in the serum from human insulin knock-in mice (Fig 2D). PCR analysis revealed that human insulin, but not murine insulin 1 encoding mRNA, could be detected in pancreatic extracts from HuPI mice (Fig 2C and S4 Table). In contrast, murine insulin 2 and beta actin mRNA were detected in pancreatic extracts from both WT and HuPI mice.

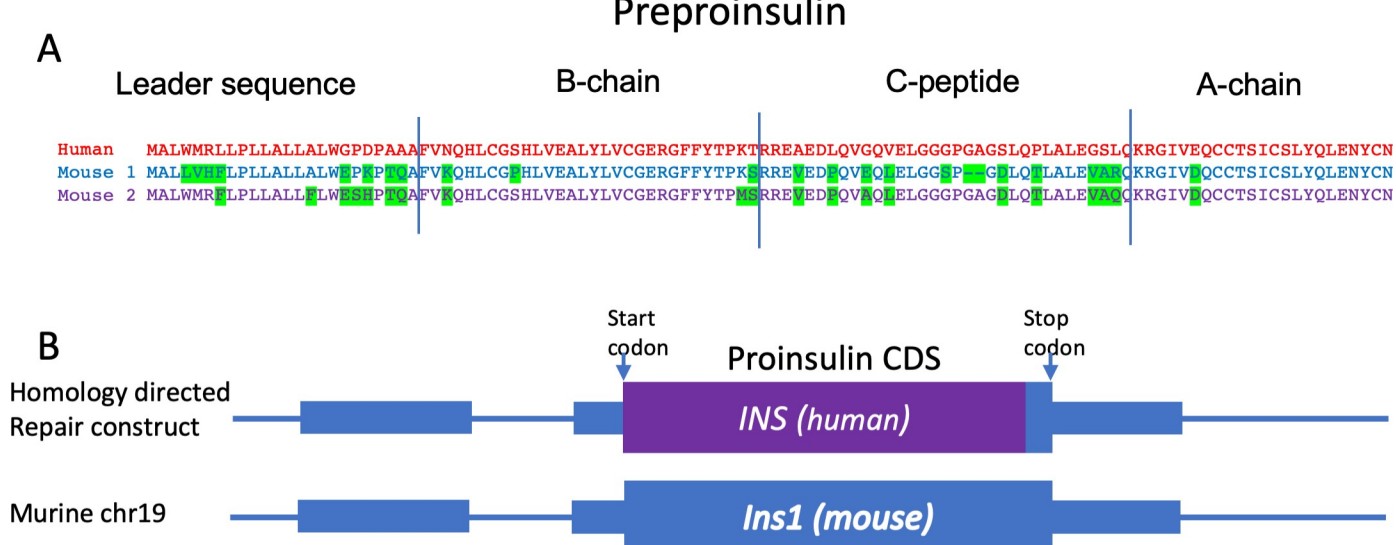

**Fig 1. Replacement of the murine *Ins1* gene with human *INS*.** (A) Alignment of the amino acid sequence of murine and human insulin. Amino acids in murine insulin that differ from the human sequence are highlighted. (B) A schematic diagram of the replacement of the coding sequence for murine *Ins1* with human *INS*. Murine sequences are shown in blue and boxes represent exons. The translation start and stop codons are indicated by arrows. A homology directed repair (HDR) construct containing the human *INS* transcript (purple box), flanked by sequence homologous to the regions flanking murine *Ins1* was used to introduce *INS* by CRISPR-Cas9 mediated targeting to the *Ins1* locus. (5' homology: 701bp; 3' homology: 559bp).

Both human and murine insulin 2 are expressed in pancreatic extracts from HuPI mice. Comparison of murine and human insulin revealed that, although variable between mice, less human insulin was expressed compared to murine insulin (Fig 2D). Overall insulin production was similar; when more human insulin was expressed less murine insulin was produced. To evaluate the HuPI mice's ability to metabolize glucose, a glucose tolerance test was performed. No difference in glucose tolerance was observed between NOD and HuPI mice (Fig 2E). To confirm that human insulin is expressed in pancreatic beta cell we analyzed insulin expression by immunohistochemistry. Human insulin was detected in beta cells from HuPI mice (Figs 3 and S2). We conclude that the HuPI mouse expresses murine insulin 2 and human insulin in place of mouse insulin 1.

## HuPI mice have a reduced incidence of diabetes

Human insulin knock-in mice, both homozygous and heterozygous, had a significantly reduced incidence of diabetes (p<0.0001, Log-rank test) compared to wildtype NOD mice when cohorts were followed for 300 days (Fig 4A). To assess the level of islet infiltration, pancreata were collected from HuPI homozygous, heterozygous and wildtype mice at approximately 100 days of age. At this time islet infiltration is commonly seen in wildtype NOD mice. Homozygous and heterozygous human insulin knock-in mice exhibited reduced insulitis at 100 days (Figs 4B and 4C and S3) compared to wildtype NOD mice. By 300 d, the insulitis in the HuPI heterozygous and homozygous was equal to wild type NOD mice (Figs 4B and 4C and S4).

## Discussion

Here we describe the generation and characterization of a NOD mouse that expresses human insulin in place of murine insulin 1. We show that human insulin is expressed in the pancreatic

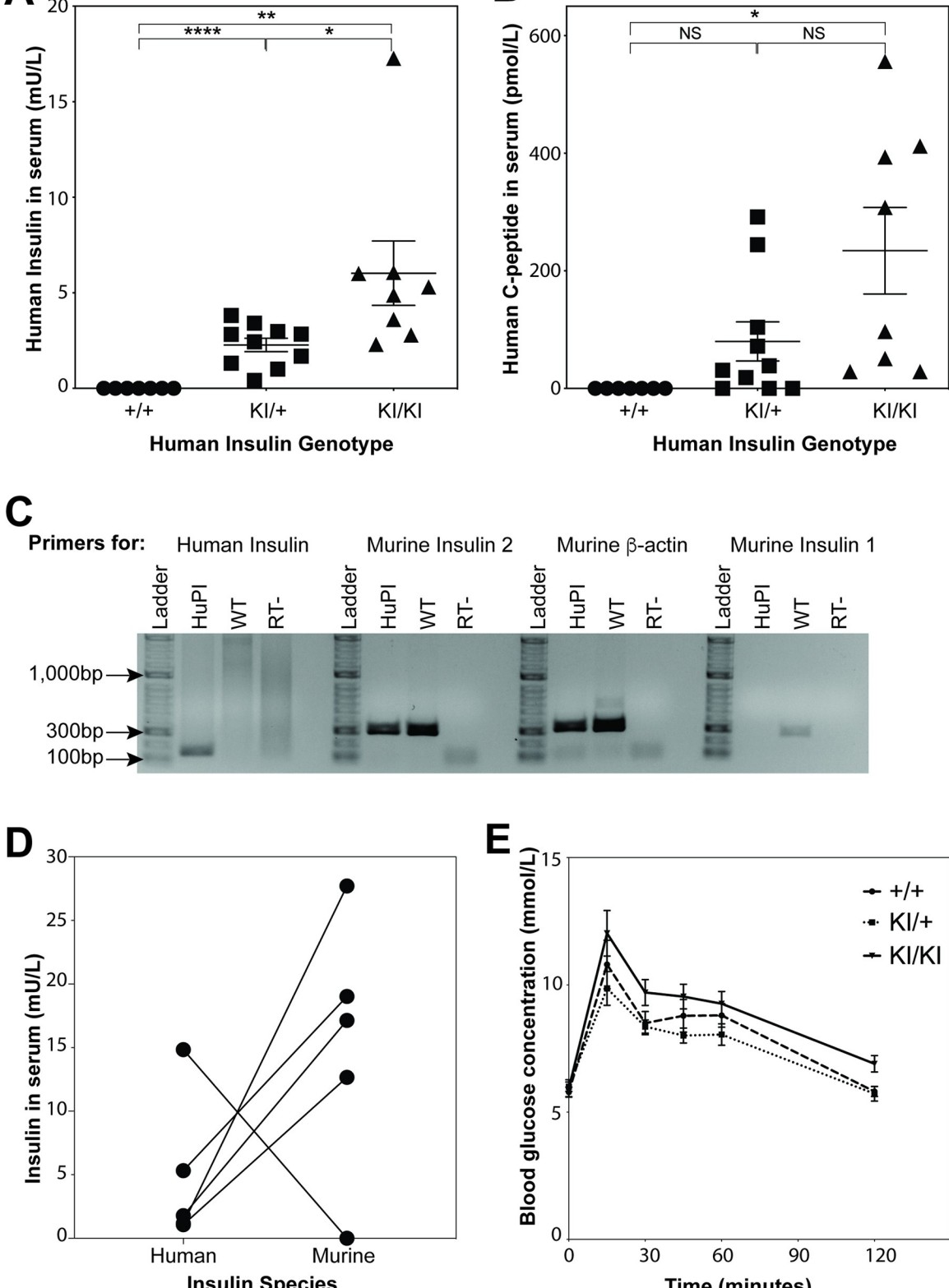

**Fig 2. Human insulin and C-peptide can be detected in NOD.HuPI serum and NOD.HuPI mice have normal glucose tolerance.**
ELISA was used to measure (A) human insulin and (B) human C-peptide concentrations in serum from fasted NOD.HuPI wildtype

(+/+, n = 7), heterozygous knock-in (KI/+, n = 10) and homozygous knock-in (KI/KI, n = 8) mice. Data are expressed as mean ± SEM. NS: p > 0.05, * p < 0.05, ** p < 0.01, **** p < 0.0001; unpaired Student's two-tailed t test. (C) Detection of cDNA specific for human insulin (*INS*), murine insulin 1 (*Ins1*) and murine insulin 2 (*Ins2*) in pancreatic cells. (D) Human and mouse insulin were measured in serum from homozygous knock-in mice (n = 5). (E) A glucose tolerance test was performed on fasted female NOD.HuPI wildtype (+/+, n = 8), heterozygous knock-in (KI/+, n = 8) and homozygous knock-in (KI/KI, n = 8) mice. p > 0.05; two-way ANOVA.

islets and human insulin and C-peptide is present in the serum. These mice breed normally, have normal glucose tolerance and are largely protected from diabetes.

Genetic modification by CRISPR/Cas9 raises the possibility that the phenotype observed is attributable to off-target genomic mutations. We believe that this is very unlikely for the following reasons. The founder lines were backcrossed to NOD for two, or in most cases, three generations. This reduces the load of putative off target mutations by >87%. No differences were noticed between mice in different branches of the pedigree, or with numbers of backcrosses. In addition, most controls were littermates of the mice harboring the HuPI insertion. The sgRNAs used for the CRISPR/Cas9 targeting resulting in the founder mutation had moderate off target scores of 47 and 67 (according to IDT's algorithm: https://sg.idtdna.com/site/order/designtool/index/CRISPR_SEQUENCE) suggesting that they had a modest propensity to make off target modifications.

As expected, because the murine insulin 1 gene regulatory elements were left intact, human insulin expression was restricted to the islets of Langerhans. Human insulin and C-peptide were detected in the serum, suggesting that human proinsulin was processed to insulin and C-peptide by murine pancreatic beta cells. The HuPI mice showed no deficit in glucose tolerance, indicating that the combined levels of human insulin and murine insulin 2 were sufficient to maintain normal glucose regulation.

Surprisingly HuPI mice were protected from spontaneous diabetes. This is similar to the phenotype of murine *Ins1* knock-out NOD mice [20]. Moriyama et al [20] found that approximately 10% more $Ins1^{+/-}$ mice developed diabetes than $Ins1^{-/-}$ mice at 300, whereas we did not

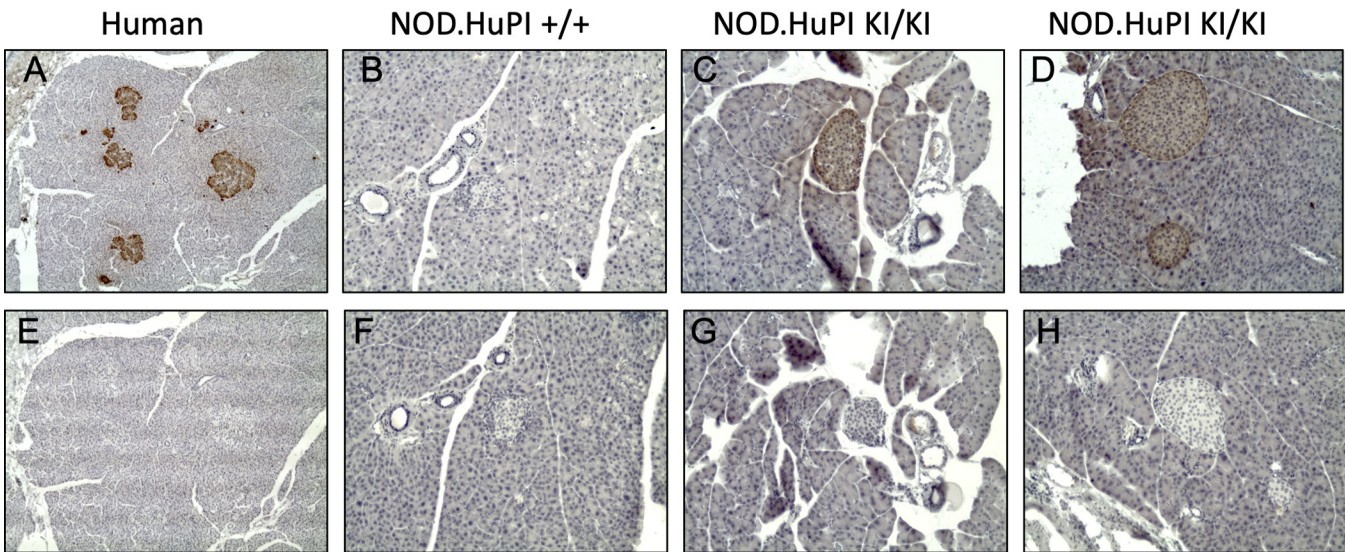

**Fig 3. Human insulin is localised to the islets of NOD.HuPI mice.** Pancreas sections from human (A, E), NOD.HuPI wildtype (B, F), and homozygous knock-in (C, D, G, H) were stained with anti-human proinsulin (GN-ID4) (A–D, top row), or an isotype control antibody (40BI3.2.1-s) (E–H, bottom row). Positive staining is indicated by the brown color. No staining was seen in pancreas of HuPI wildtype mice. Photos taken at 100x magnification. Representative images are shown.

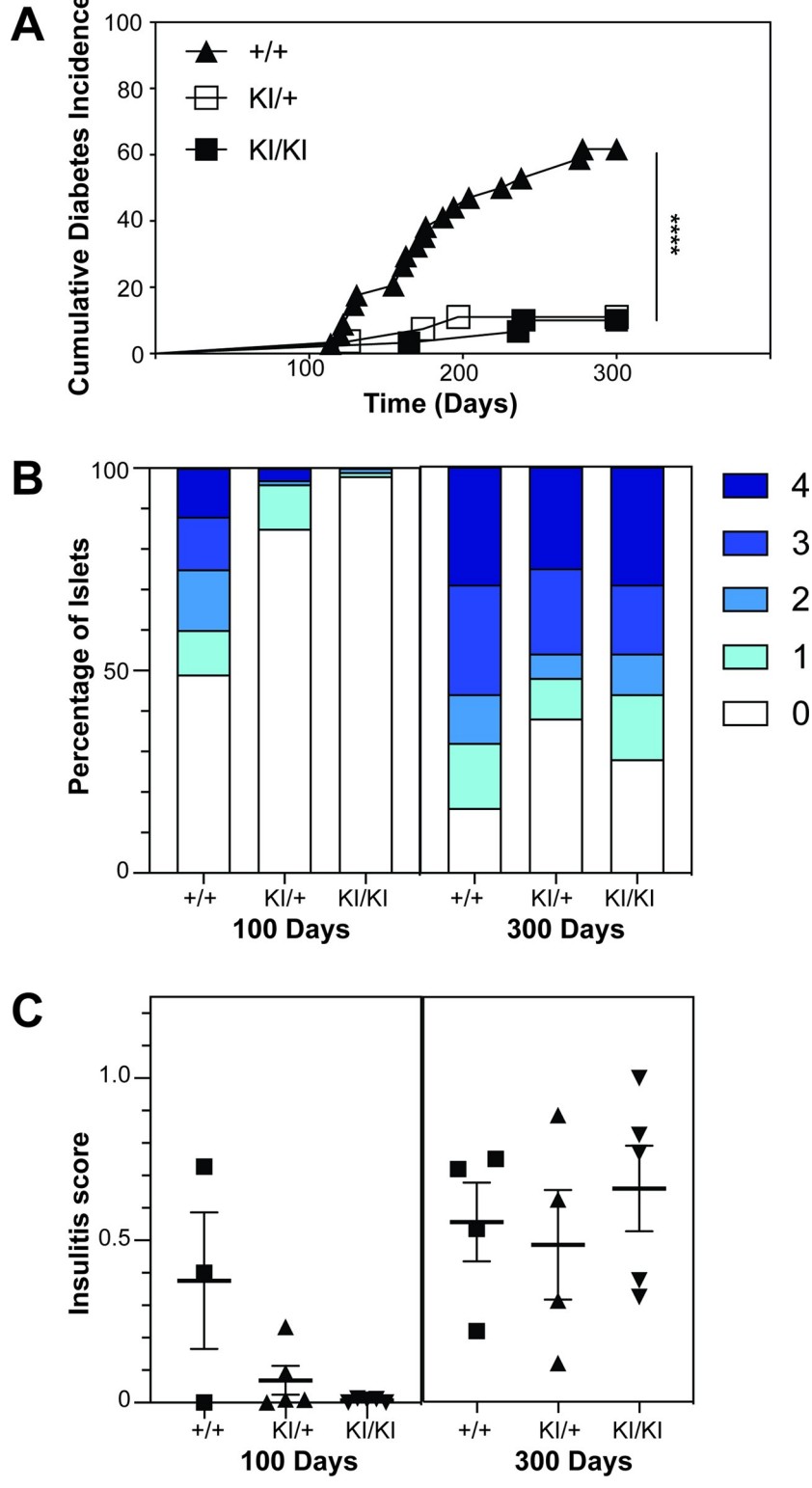

**Fig 4. NOD.HuPI mice have reduced incidence of diabetes and delayed insulitis.** (A) Female homozygous NOD.
HuPI knock-in (KI/KI, N = 30), heterozygous knock-in (KI/+, n = 27) or wildtype (+/+, n = 34) mice were aged for
300 days. Urinary glucose was tested weekly. Mice were declared diabetic if they had high urinary glucose readings for
three consecutive days and a blood glucose reading >15mmol/L. **** p < 0.0001 Log-rank test +/+ vs KI/KI and
+/+ vs KI/+. Lymphocytic infiltration of the islets was scored in haematoxylin and eosin-stained pancreas sections of

female NOD.HuPI homozygous knock-in (KI/KI, n = 5), heterozygous knock-in (KI/+, n = 5) and wildtype (+/+, n = 3) 100d-old mice and female NOD.HuPI homozygous knock-in (KI/KI, n = 5), heterozygous knock-in (KI/+, n = 5) and wildtype (+/+, n = 4) 300d-old mice (B, C). Islet infiltration scoring: 0 = no infiltration, 1 = peri-insulitis, 2 = < 25% islet infiltrated, 3 = > 25% of the islet infiltrated and 4 = complete infiltration. A weighted average insulitis score was calculated as described in the methods. 100d p < 0.05; 300d p > 0.05; one-way ANOVA.

see any difference in diabetes incidence between HuPI homozygous or heterozygous mice at this time (Fig 4A).

Currently it is not clear how human insulin protects NOD mice from diabetes. One possibility is that the NOD mouse repertoire is 'blind' to human insulin, despite the similarity in the protein sequence between human insulin and murine insulin 1 (Fig 1). However, this does not account for the protection from diabetes in the HuPI heterozygous mice. In the heterozygous mice the concentration of *Ins1* may be below a critical threshold required for the onset of diabetes. It should be noted that HuPI mice were not completely protected from diabetes, but the incidence of disease was significantly reduced with a delayed onset. Histological analysis showed that there was little islet infiltration at 100 days, but this increased markedly by 300 days. Our incidence studies were stopped after 300 days of age however it is possible that the incidence of diabetes in the HuPI mice would increase if they were monitored for longer.

The amino acid sequences are identical between mouse and human proinsulin in the insulin B9-23 epitope (Fig 1A). NOD mice that carried a tyrosine (Y) to alanine (A) mutation position B16 are protected from diabetes [9]. Most of the amino acid differences between mouse and human proinsulin fall within the C-peptide. NOD mouse CD4 epitopes have been described from this region [32]. This suggests that changes in the sequence of proinsulin C-peptide may contribute to the low incidence of diabetes in HuPI mice. Recently we reported that full-length C-peptide is a major antigen in human type 1 diabetes [14]. Our current observations suggest it may also play a greater role in the NOD mouse than previously appreciated.

HuPI mice still express murine insulin 2. Human insulin was readily detected in the serum and islets of HuPI mice and mRNA for murine insulin 1 was undetectable. It will be of interest to generate NOD mice with both murine insulin 1 and insulin 2 replaced with human insulin, which would only make human insulin. Because proinsulin, the precursor of insulin, is the major protein product of beta cells, accounting for ~10% of the cell's protein [33], it is possible that a fully human insulin knock-in mouse may have dysfunctional beta cells due to misfolding of human insulin in a murine beta cell. However, our current results suggest that a mouse that only expresses human insulin would be healthy. Murine insulin knockout mice that express human insulin from a transgene have been generated and appear to have no metabolic abnormalities [34].

To develop a murine model of human autoimmune responses that cause type 1 diabetes will require further modification of the HuPI mouse. Such a model will need to express T1D associated HLA alleles, namely HLA-DQ2, DQ8 (and their transdimers), HLA-DR4 or HLA-DR3 [35]. In addition, a TCR specific for human proinsulin, or other relevant beta-cell antigens, will be required. It will be a high priority to investigate the function of TCRs isolated from human islet infiltrating T cells [12], because these cells are most likely to be pathogenic [1]. The 'Yes' mouse is, to our knowledge, the most advanced transgenic model for human T1D [34]. This mouse expresses HLA-A2:01, HLA-DQ8 and carries human insulin as transgenes, with the murine orthologues of these genes knocked out. This model is on a mixed C57BL/6 and CBA background. One advantage of CRISPR/Cas9 mediated gene replacement is that it can be done on a NOD background, which avoids issues related to mixed genetics. Verhagen et al [27] recently reported an HLA-DR4 transgenic mouse that expresses CD80 on its pancreatic beta cells. Diabetes can be induced in this mouse by immunization with murine

insulin 2 derived peptides in adjuvant. While these mice can be manipulated to develop diabetes, it is unclear how faithfully this model recapitulates the human disease.

At this stage it is unknown if human proinsulin specific TCRs, or T cells, will respond to human insulin expressed by the HuPI mice. Addressing this question will require a NOD mouse that expresses human CD4, (pro)insulin specific TCR and the restricting HLA allomorph for the T cell to be generated. Nonetheless, the feasibility of using CRISPR/Cas9 to replace murine genes with their human orthologues augers well for a new generation of mouse models for T1D and other human autoimmune diseases. Recent advances in using CRISPR/Cas9 to integrate large constructs into the murine genome will greatly assist in generating these models [36].

## Supporting information

**S1 Fig. Sequence of the homology-directed repair (HDR) template.**
(PDF)

**S2 Fig. Human insulin is localized to the pancreatic islets in NOD.HuPI mice.**
(PDF)

**S3 Fig. NOD.HuPI mice exhibit delayed insulitis (100d).**
(PDF)

**S4 Fig. NOD.HuPI mice exhibit delayed insulitis (300d).**
(PDF)

**S1 Table. Production of human insulin knock-in mice by CRISPR/Cas9 mutagenesis.**
(PDF)

**S2 Table. Molecular confirmation of *INS* knock-in.**
(PDF)

**S3 Table. Genotype ratios in offspring of NOD.HuPI mice.**
(PDF)

**S4 Table. Insulin-specific primers.**
(PDF)

**S1 Raw Image. Original gel image of Fig 2C.**
(TIF)

## Acknowledgments

The mutant mice were produced via CRISPR/Cas9 oocyte injection by Monash University as a node of the Australian Phenomics Network (APN). The APN is supported by the Australian Government Department of Education through the National Collaborative Research Infrastructure Strategy, the Super Science Initiative and the Collaborative Research Infrastructure Scheme.

The authors thank the staff at St. Vincent's Bioresources Centre for excellent animal husbandry.

## Author Contributions

**Conceptualization:** Colleen M. Elso, Andrew P. R. Sutherland, Helen E. Thomas, Stuart I. Mannering.

**Data curation:** Colleen M. Elso, Nicholas A. Scott, Stuart I. Mannering.

**Formal analysis:** Colleen M. Elso, Nicholas A. Scott.

**Funding acquisition:** Stuart I. Mannering.

**Investigation:** Colleen M. Elso, Nicholas A. Scott, Lina Mariana, Emma I. Masterman.

**Methodology:** Colleen M. Elso, Nicholas A. Scott, Lina Mariana, Emma I. Masterman, Andrew P. R. Sutherland.

**Project administration:** Colleen M. Elso, Helen E. Thomas, Stuart I. Mannering.

**Resources:** Helen E. Thomas.

**Supervision:** Helen E. Thomas, Stuart I. Mannering.

**Writing – original draft:** Stuart I. Mannering.

**Writing – review & editing:** Colleen M. Elso, Stuart I. Mannering.

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
