## [Decision Letter · Decision Letter 0]

11 Nov 2019

PONE-D-19-29642

Replacing murine insulin 1 with human insulin protects NOD mice from diabetes

PLOS ONE

Dear A/Prof Mannering,

Thank you for submitting your manuscript to PLOS ONE. After careful consideration, we feel that it has merit but does not fully meet PLOS ONE’s publication criteria as it currently stands. Therefore, we invite you to submit a revised version of the manuscript that addresses the points raised during the review process.

ACADEMIC EDITOR: I like this paper  my main concern would be about off target effects /integrations ?  Matthias 

We would appreciate receiving your revised manuscript by Dec 26 2019 11:59PM. To enhance the reproducibility of your results, we recommend that if applicable you deposit your laboratory protocols in protocols.io, where a protocol can be assigned its own identifier (DOI) such that it can be cited independently in the future. For instructions see: http://journals.plos.org/plosone/s/submission-guidelines#loc-laboratory-protocols

We look forward to receiving your revised manuscript.

Kind regards,

Matthias G von Herrath, MD PhD

Academic Editor

PLOS ONE

Additional Editor Comments (if provided):

Overall I think this is a very interesting paper - my main concern would indeed relate to the issue whether there were any off targeted integrations? Disruptions? Matthias

Journal Requirements:

2) PLOS ONE now requires that authors provide the original uncropped and unadjusted images underlying all blot or gel results reported in a submission’s figures or Supporting Information files. This policy and the journal’s other requirements for blot/gel reporting and figure preparation are described in detail at https://journals.plos.org/plosone/s/figures#loc-blot-and-gel-reporting-requirements and https://journals.plos.org/plosone/s/figures#loc-preparing-figures-from-image-files. When you submit your revised manuscript, please ensure that your figures adhere fully to these guidelines and provide the original underlying images for all blot or gel data reported in your submission. See the following link for instructions on providing the original image data: https://journals.plos.org/plosone/s/figures#loc-original-images-for-blots-and-gels.

Reviewers' comments:

Reviewer's Responses to Questions

**Comments to the Author**

1. Is the manuscript technically sound, and do the data support the conclusions?

Reviewer #1: Partly

Reviewer #2: Yes

2. Has the statistical analysis been performed appropriately and rigorously? 

Reviewer #1: N/A

Reviewer #2: Yes

3. Have the authors made all data underlying the findings in their manuscript fully available?

Reviewer #1: Yes

Reviewer #2: Yes

4. Is the manuscript presented in an intelligible fashion and written in standard English?

Reviewer #1: Yes

Reviewer #2: Yes

5. Review Comments to the Author

Reviewer #1: This is an observational study where the authors have replaced the mouse INS1 gene with the human INS sequence and found that the resulting trangenic strain is protected against spontaneous autoimmune diabetes development. No mechnistic studies were performed so it is unclear why the mice are protected.

The key unknown in this story is whether the INS1/hINS replacement is indeed the (sole) cause of the reduced disease penetrance. CRISPR-Cas9 mediated double break induction and HDR is notoriously non-specific. Since no off-target analysis was performed at all, how certain are the authors that none of the breaks/insertions landed in e.g. an immune gene?

What do the authors mean by 'most of the amino acid coding region' being replaced (l142, p8). The Figure is too low resolution to check for myself.

The anitbodies used for ELISA and IHC to detect mouse vs human insulin: how certain are the authors that there is no cross-reactivity?

Reviewer #2: In this manuscript, entitled “Replacing murine insulin 1 with human insulin protects NOD mice from diabetes”, the authors describe the generation of a novel human insulin-expressing NOD mouse strain. To achieve this, the authors replaced the murine insulin 1 gene ( Ins1 ) in NOD mice with the human insulin gene ( Ins ) using Crispr/Cas9 technology. These mice (HuPI) express human insulin in place of murine insulin 1 in their islets, and human C-peptide can be detected in their serum. These mice show also lower incidence of T1D as compared to wild-type NOD mice. Undoubtedly, this can be a useful novel humanized model of T1D. Unfortunately, the study lacks immunological and overall mechanistic insights. Below we suggest a few, relatively simple experiments.

Comments/Suggestions

1. How do the authors explain the fact that KI/+ mice have the same incidence of T1D as KI/KI mice?

2. The authors need to discuss more the similarities and differences in the amino acid sequence between mouse ins1, ins2 and human insulin, particularly with regards the immunodominant epitopes, for example B:9-23. It would be of value to discuss Maki Nakayama’s paper in Nature 2004 within this context.

3. The authors need to measure mouse ins1 and 2 as well human insulin in the thymus of +/+, KI/+ and KI/KI mice.

4. Statistics are missing from the M&M section.

5. We believe that the authors should back-up the protein expression data of human insulin with quantitative RNA analyses as well.

6. Could the authors purify splenocytes from diabetic +/+, KI/+ and KI/KI mice and transfer them to NOD.Scid to further evidence the autoreactive nature of T cells?

7. Could they evaluate Treg frequency and numbers and perhaps perform Treg transfers from KI/KI mice to +/+?

8. What do the authors think will be the phenotype of KI mice crossed to ins2-deficient background? Are they planning on it? Could they discuss it?

9. Could they discuss other similar models (ref 33) further and perhaps how their model could be improved, for example by expressing human HLA?

10. It is not clear how the PCR analyses were done (lines 168-170).

6. PLOS authors have the option to publish the peer review history of their article (what does this mean?). If published, this will include your full peer review and any attached files.

Reviewer #1: No

Reviewer #2: No

---

## [Author Response · Author response to Decision Letter 0]

20 Nov 2019

RE: PONE-D-19-29642 

“Replacing murine insulin with human insulin protects NOD mice from diabetes”

Response to the reviewers’ comments

Dear Editor,

First, we’d like to thank the Academic Editor and the reviewers for their constructive and thoughtful comments on this manuscript. Below, we address each of the points raised. The person/people who made these comments is/are show in in parenthesis. The comment, or a precis of it, is shown in bold and our response in plain text. The changes to the manuscript text are highlighted in red text on the tracked change version.

1. (Academic Editor and Reviewer 1) Are there any off-target effects of the CRISPR/Cas9? 

We are confident that the protection that we observe in NOD mice is not attributable to off-target genomic mutations caused by CRISPR/Cas 9 for the following reasons. 

1. The mutation was back crossed to NOD mice for three generations. Each back cross reduces the load of putative off target mutations by 50%, so after three generations >87% of putative off-target mutations would have been lost.

2. There was no difference in the diabetes incidence of mice from different branches of the pedigree, suggesting that the phenotype was due to the alteration of the INS1 locus not elsewhere in the genome. We saw no difference between litters; if putative mutations were segregating and responsible for the phenotype, we would have expected to see differences between litters which we did not. The wildtype mice were a mix of littermate controls and NOD mice from our colony. There were no differences between these two groups of control mice. If there was an off-target or background effect, then we would have seen it there too. 

3. Off-target effects of CRISPR have been reported to be lower in vivo than similar experiments in vitro [1]. Furthermore, the putative CRISPR/Cas9 introduced mutations should be considered in the context of the background mutation rate per mammalian generation has been estimated to be 70-175 de novo mutations per diploid genome per generation [2, 3].

4. The single guide RNAs had relatively high off-target scores on IDT’s algorithm (https://sg.idtdna.com/site/order/designtool/index/CRISPR_SEQUENCE). The sgRNAs used that led to the mouse in the study had off target scores of 47 and 67, indicating that they have a modest risk of off target effects. These scores are now included in Supplementary Table 1B.

5. In addition to the points raised above and given insulin’s well-established role in the immune pathogenesis of diabetes, both in humans and NOD mice it is extremely unlikely that protection from diabetes would develop in CRISPR-modified mice due to a mutation in an unrelated gene.

The possible role of off target mutations is now discussed on page 5, lines 97-100 and in the Discussion (page 13, lines 256-265).

2.(Reviewer 1) What do the authors mean by 'most of the amino acid coding region' being replaced (l142, p8)?

Because the final 16 amino acids in the A-chain of murine insulin 1 and human insulin are identical this part of the gene was not modified. This has now been clarified on page 9, lines 163-165.

3. (Reviewer 1) The antibodies used for ELISA and IHC to detect mouse vs human insulin: how certain are the authors that there is no cross-reactivity?

We have used well established and characterized mAbs and appropriate controls to exclude the possibility of any cross reactivity between mouse and human (pro)insulin. 

1. For IHC, the mAb used for histology has been extensively characterized and shown to be specific for human insulin [4]. Furthermore, we did not detect any staining in sections from mice that did not express human insulin.

2. For ELISAs we used a commercial ELISA kit from Mercodia ELISAs which is specific for human insulin. The mAbs used in the ELISA assays for mouse insulin do cross react with human and murine insulin. However, human insulin can be measured without cross reactivity with murine insulin. To determine the concentration of murine insulin the human insulin component is subtracted, according to the manufacturer’s protocol. This is now described more clearly in Methods section.

4.(Reviewer 2) How do the authors explain the fact that KI/+ mice have the same incidence of T1D as KI/KI mice?

The simplest interpretation of this observation is that the expression of human insulin, in place of murine insulin 1 induces dominant protection against autoimmune diabetes. At this stage it is not known whether this is mediated by a Treg population, or impacts upon thymic selection, or another pathway. A thorough analysis of the mechanism(s) underlying the reduced incidence of autoimmune diabetes in this model would require several years work and is therefore beyond the scope of the current manuscript.

5. Discuss more the similarities and differences in the amino acid sequence between mouse ins1, ins2 and human insulin, with regards B:9-23 and Maki Nakayama’s paper in Nature 2004

The amino acid sequence of insulin B9-23 epitope is identical between murine insulin 1, murine insulin 2 and human insulin (see Figure 1). All have the pivotal B16 tyrosine residue that was mutated to alanine in Dr Nakayama’s 2005 Nature paper (Ref No 9 in the manuscript). Figure 1 highlights that the most difference in amino acid sequence are found in the C-peptide of proinsulin. This is consistent with reports that this region, in addition to B9-23, harbors epitopes important for the immune mediated destruction of beta cells in the NOD mouse model [5]. This is now discussed in the Discussion (p14-15, lines 300-308).

6. The authors need to measure mouse ins1 and 2 as well human insulin in the thymus of +/+, KI/+ and KI/KI mice.

As noted above a thorough analysis of the mechanism of diabetes suppression in this model is beyond the scope of the current manuscript. From our genetic analysis we know that the HuPI KI mice have human insulin in place of murine insulin 1. For this reason, KI/KI mice will not have any murine insulin 1 in the thymus. We have not modified the murine insulin 2 locus, so this will not be altered. We assume that human insulin, like murine insulin 1, will be almost exclusively detected in the pancreatic beta cells consistent with the data we have presented.

7. (Reviewer 2) Statistics are missing from the M&M section

Thank you for drawing this to our attention, we have now included a description of the statistical analysis into the relevant parts of the Methods section.

8. (Reviewer 2) The authors should back-up the protein expression data of human insulin with quantitative RNA analyses

With respect, we disagree. We have already shown that mRNA for human insulin is expressed in the islets of the HuPI mice (Figure 2C). We have also shown that both human insulin protein and C-peptide are expressed. In our view, the critical point is that human insulin and C-peptide are expressed, albeit at variable concentrations. In our view measuring the mRNA levels will not give further insights above what have already be reported.

9. (Reviewer 2) Could the authors:

Purify splenocytes from diabetic +/+, KI/+ and KI/KI mice and transfer them to NOD.Scid?

Evaluate Treg frequency and numbers and perhaps perform Treg transfers from KI/KI mice to +/+?

We thank Reviewer 2 for these helpful suggestions. We are planning a manuscript which will address the mechanism(s) by which HuPI mice are protected from diabetes. However, to come to a clear conclusion regarding the mechanism is beyond the scope of this manuscript. Here our aim is to report the generation and evaluation of HuPI mice. 

10. Could they discuss other similar models (ref 33) further and perhaps how their model could be improved, for example by expressing human HLA?

A more detailed discussion of other models has now been included in the Discussion (p15-16 line 321-338). We also discuss the further development of this model, including the expression of HLA and human (pro)insulin specific TCRs.

11. It is not clear how the PCR analyses were done (lines 168-170).

Thank you for drawing this to our attention. In addition to the primers’ sequences we have now included more details about how the PCR was performed in the Materials and Methods section of the manuscript (p6, line 113-120).

References cited

1. Yoshimi K, Kaneko T, Voigt B, Mashimo T. Allele-specific genome editing and correction of disease-associated phenotypes in rats using the CRISPR-Cas platform. Nature communications. 2014;5:4240. Epub 2014/06/27. doi: 10.1038/ncomms5240. PubMed PMID: 24967838; PubMed Central PMCID: PMCPMC4083438.

2. Nachman MW, Crowell SL. Estimate of the mutation rate per nucleotide in humans. Genetics. 2000;156(1):297-304. Epub 2000/09/09. PubMed PMID: 10978293; PubMed Central PMCID: PMCPMC1461236.

3. Roach JC, Glusman G, Smit AF, Huff CD, Hubley R, Shannon PT, et al. Analysis of genetic inheritance in a family quartet by whole-genome sequencing. Science. 2010;328(5978):636-9. Epub 2010/03/12. doi: 10.1126/science.1186802. PubMed PMID: 20220176; PubMed Central PMCID: PMCPMC3037280.

4. Asadi A, Bruin JE, Kieffer TJ. Characterization of Antibodies to Products of Proinsulin Processing Using Immunofluorescence Staining of Pancreas in Multiple Species. J Histochem Cytochem. 2015;63(8):646-62. Epub 2015/07/29. doi: 10.1369/0022155415576541. PubMed PMID: 26216140; PubMed Central PMCID: PMCPMC4530395.

5. Levisetti MG, Lewis DM, Suri A, Unanue ER. Weak Proinsulin Peptide–Major Histocompatibility Complexes Are Targeted in Autoimmune Diabetes in Mice. Diabetes. 2008;57(7):1852-60. doi: 10.2337/db08-0068.

---

## [Editor Report · Decision Letter 1]

22 Nov 2019

Replacing murine insulin 1 with human insulin protects NOD mice from diabetes

PONE-D-19-29642R1

Dear Dr. Mannering,

We are pleased to inform you that your manuscript has been judged scientifically suitable for publication and will be formally accepted for publication once it complies with all outstanding technical requirements.

With kind regards,

Matthias G von Herrath, MD PhD

Academic Editor

PLOS ONE

Additional Editor Comments (optional):

Good paper! :-)
---

## [Editor Report · Acceptance letter]

2 Dec 2019

PONE-D-19-29642R1 

Replacing murine insulin 1 with human insulin protects NOD mice from diabetes 

Dear Dr. Mannering:

I am pleased to inform you that your manuscript has been deemed suitable for publication in PLOS ONE. Congratulations! Your manuscript is now with our production department. 

With kind regards,

on behalf of

Prof. Matthias G von Herrath 

Academic Editor

PLOS ONE